# The Influence of Dietary Gallic Acid on Growth Performance and Plasma Antioxidant Status of High and Low Weaning Weight Piglets

**DOI:** 10.3390/ani11113323

**Published:** 2021-11-21

**Authors:** Xuemei Zhao, Jizhe Wang, Ge Gao, Valentino Bontempo, Chiqing Chen, Martine Schroyen, Xilong Li, Xianren Jiang

**Affiliations:** 1Key Laboratory of Feed Biotechnology of the Ministry of Agriculture, Institute of Feed Research, Chinese Academy of Agricultural Sciences, Beijing 100081, China; xuemei.zhao@doct.uliege.be (X.Z.); 82101196174@caas.cn (J.W.); 82101192355@caas.cn (G.G.); lixilong@caas.cn (X.L.); 2TERRA Teaching and Research Centre, Precision Livestock and Nutrition Laboratory, Gembloux Agro-Bio Tech, University of Liège, 5030 Gembloux, Belgium; martine.schroyen@uliege.be; 3Department of Health, Animal Science and Food Safety, Università degli Studi di Milano, Via dell’Università 6, 26900 Lodi, Italy; valentino.bontempo@unimi.it; 4Wufeng Chicheng Biotech Co., Ltd., Yichang 443413, China; wfcchem@126.com

**Keywords:** gallic acid, growth performance, diarrhea incidence, antioxidant capacity

## Abstract

**Simple Summary:**

Gallic acid (GA) has been demonstrated to have antioxidant, antimicrobial, anti-inflammatory, and health-promoting properties. In pigs, GA supplementation has been shown to decrease di-arrhea incidence of weaned piglets and improve their intestinal integrity. The present experiment was conducted to test the hypothesis that growth performance and diarrhea after weaning could be improved by supplementing the diet with 400 mg/kg GA to weaned piglets, especially for low weaning weight piglets.

**Abstract:**

This study evaluated the effects of dietary gallic acid (GA) on growth performance, diarrhea incidence and plasma antioxidant status of weaned piglets regardless of whether weaning weight was high or low. A total of 120 weaned piglets were randomly allocated to four treatments in a 42-day experiment with a 2 × 2 factorial treatment arrangement comparing different weaning weights (high weight (HW) or low weight (LW), 8.49 ± 0.18 kg vs. 5.45 ± 0.13 kg) and dietary treatment (without supplementation (CT) or with supplementation of 400 mg/kg of GA). The results showed that HW piglets exhibited better growth performance and plasma antioxidant capacity. Piglets supplemented with GA had higher body weight (BW) on day 42 and average daily gain (ADG) from day 0 to 42 compared to the control piglets, which is mainly attributed to the specific improvement on BW and ADG of LW piglets by the supplementation of GA. The decreased values of diarrhea incidence were seen in piglets fed GA, more particularly in LW piglets. In addition, dietary GA numerically reduced malondialdehyde (MDA) content in plasma of LW piglets. In conclusion, our study suggests that dietary GA may especially improve the growth and health in LW weaned piglets.

## 1. Introduction

Gallic acid (GA) is a well-known endogenous plant polyphenol present in fruits, nuts, and plants [1,2,3]. As a natural antioxidant, GA prevents the damage induced by reactive oxygen species (ROS) [4] mainly via the scavenging effect on hydroxyl radical and hydrogen peroxides [5]. Next to its antioxidant effect, GA also inhibits the motility, adherence and biofilm formation of bacteria [6,7], accelerates the accumulation of antibiotics in microorganisms [8], and therefore exhibits antimicrobial effects. In addition, GA not only modulates the function of basophils and reduces the release of histamine, but also suppresses the production of pro-inflammatory cytokines in macrophages. Due to the antioxidant, antimicrobial, anti-inflammatory, and health-promoting effects, GA has been extensively studied as feed supplementation in animal production. Chickens fed diets supplemented with GA at 75 to 100 mg/kg displayed a promotion in growth and feed utilization, the integrity and morphology of jejunum were positively modulated [9]. Diets with GA (400 mg/kg) decreased postweaning diarrhea and protected intestinal integrity in pigs [10]. Due to the association interactions with water [11] and the rapid absorption in the stomach and small intestine of animals [12], GA has also shown a higher bioavailability. 4-O-methylgallic acid (the main derivative of GA), free, and glucuronidated forms of gallic acid are the main metabolites of GA in blood of animals and humans [13,14]. 

Weaning is one of the most stressful events for piglets due to the sudden changes in physiological status and environment. Piglets easily experience low feed intake and an increased prevalence of diarrhea, which have negative effects on growth performance [15]. In addition, weaning also induces oxidative stress, which has negative effects on piglets’ health [16].Moreover, low weight has been associated with lower immune development and a higher prevalence of diseases [17]. 

Our previous study has shown that diet supplemented with GA at 400 mg/kg decreased diarrhea incidence of weaned piglets with an average weaning weight at 8.40 ± 0.09 kg and improved their intestinal morphology [10]. However, the positive effect on growth performance for weaned piglets was not observed. Therefore, we hypothesized that dietary GA supplementation at 400 mg/kg could improve growth performance and decrease diarrhea incidence for weaned piglets, especially for low weaning weight piglets. The aim of the study was to evaluate the effect of dietary GA supplemented to weaned piglets on their performance and diarrhea incidence after weaning. In addition, the goal was to investigate if supplementation of GA in weaned piglet’s diet would improve antioxidant capacity.

## 2. Materials and Methods

The experimental protocol was approved by the Animal Care and Use Committee of the Chinese Academy of Agricultural Sciences with an approved number FRI-CAAS-20200815. 

### 2.1. Animals and Experimental Design

The experiment was arranged as a 2 × 2 factorial study. The factors evaluated were weaning weight [high weight (HW) or low weight (LW)] and dietary treatment [control, without supplementation (CT) or supplementation with 400 mg/kg of gallic acid (GA)]. The research was conducted at the Tianpeng husbandry located at Langfang, Hebei province. The GA used in the present experiment was provided by Wufeng Chicheng Biotech Co., Ltd. (Yichang, China). A total of 120 crossbred (Duroc × Landrace × Yorkshire) piglets were weaned at 24 days of age containing 30 gilts and 30 barrows with high weaning weight (8.49 ± 0.18 kg) and 30 gilts and 30 barrows with low weaning weight (5.45 ± 0.13 kg) from the same batch of 319 piglets. All the selected piglets were assigned randomly according to sex and body weight (BW) to 4 treatments that were allocated to six replicates of each treatment. Each replicate consisted of 5 piglets that were housed in pens. The piglets were raised 42 days in 4 different treatments in a 2 × 2 factorial treatment arrangement comparing weaning weight (HW, LW) and diets (without GA (CT) or with 400 mg/kg of GA (HWCT, HWGA, LWCT, LWGA). The feeding protocol was carried out from day 0 to 42 of weaning. Corn and soybean-based diets were prepared according to the National Research Council 2012 nutrient requirements and supplemented with GA at 0 and 400 mg/kg, respectively. We added the GA to the vitamin and mineral complexes and mixed by hand, then the mixture was added to feed mixed by machine. The pre-starter period was from 0 to 14 day and starter period from 14 to 42 d of trial. On the morning of day 14 of the trial, the pre-starter feed was collected and the starter feed was administered to piglets. Piglets in HWCT and LWCT treatments were fed diets without GA, HWGA and LWGA treatments were fed diets with GA. During the trial period, all piglets had free access to food and drinking water. The temperature of the nursery house was controlled at 28 °C during the first week and was then adjusted gradually to 26 °C. Piglets were housed in a conventional nursery house where pens (2.00 × 2.00 m^2^) consisted of a slatted floor, two water nipples, and a feed trough. Diets provided during the trial were formulated according to the National Research Council (2012) nutrient requirements. Dietary phases and their duration, the composition and nutrient levels of the basal diets are shown in Table 1.

Piglets in each pen were weighed in the morning of days 0, 14, 28 and 42. The total feed consumed in each pen was recorded daily; the average daily gain (ADG), average daily feed intake (ADFI), and gain:feed ratio (G:F) were calculated every two weeks. The diarrhea incidence of each piglet was scored at the same time every morning during the first two weeks of the trial. The fecal score was based on a five-point fecal consistency scoring system: 1 = hard, dry pellet; 2 = firm, formed stool; 3 = soft, moist stool that retains its shape; 4 = soft, unformed stool; and 5 = watery liquid that can be poured. Piglets were considered to have diarrhea when the score was 4 or 5 [18,19]. The incidence of diarrhea (%) was expressed as the percentage of piglets with diarrhea in relation to the total number of weaned piglets.

### 2.2. Sample Collection

On days 14 and 42 of the trial, one piglet from each pen was selected randomly to collect blood samples from the vena jugularis externa of piglets in heparin sodium vacutainer tubes and centrifuged at 4000× *g* for 20 min. Plasma was stored at −20 °C until analysis.

### 2.3. Antioxidant Parameters Analysis

The assay kits of malondialdehyde (MDA) concentration, superoxide dismutase (SOD) activity, and glutathione peroxidase (GSH-Px) activity in plasma were purchased from Nanjing Jiancheng Bioengineering Institute. MDA concentration was determined using 2-thiobarbituric acid and the optical density (OD) value was read at 532 nm. The SOD activity was calculated through a nonenzymatic nitroblue tetrazolium (NBT) test, which measures the inhibition of the formation of superoxide anion free radicals that reduce the nitroblue tetrazolium of the sample, and the OD value was read at 450 nm. 5,50-dithiobis-p-nitrobenzoic acid was used to determine the GSH-Px activity and the OD value was read at 412 nm.

### 2.4. Statistical Analysis

The data were analyzed as a completely randomized design with a 2 × 2 factorial treatment arrangement by ANOVA using the GLM procedure in SAS v. 9.2 (SAS Inst. Inc., Cary, NC, USA). The statistical model included the effects of weaning weight (HW or LW), diet (CT or GA), and their interactions. The pen represented the experimental unit for growth performance, and the piglet was the experimental unit for plasma antioxidant. Treatment comparisons were performed using Tukey’s honestly significant difference test for multiple testing. Moreover, the chi-square test was used to analyze diarrhea incidence. Probability values of *p* ≤ 0.05 were considered to be significant, whereas a treatment effect trend was noted for *p* ≤ 0.10.

## 3. Results

### 3.1. Growth Performance and Diarrhea Incidence

The effect of dietary GA on growth performance of high and low weaning weight piglets is shown in Table 2. Piglets fed GA showed a higher BW compared to the control piglets on day 42 of the trial (*p* = 0.045). Moreover, diets with GA increased ADG from day 0 to 42 of the trial (*p* = 0.049). This increase is mainly attributed to the specific improvement on BW and ADG of LW piglets by the supplementation of GA. In addition, the interactions between weaning weight, and dietary GA showed a statistical tendency on ADFI from day 14 to 28 (*p* = 0.086) and day 28 to 42 (*p* = 0.065), respectively, which can be attributed to the difference between LWCT and LWGA, but no differences were found between HWCT and HWGA. No statistical significance was found in G:F ratio during the whole period of the trial. The effect of GA on diarrhea incidence of high and low weaning weight piglets is shown in Figure 1. Adding GA to diet decreased mean values in both HW and LW piglets (3.33% and 2.22%, respectively), although in this case, differences compared with the HWCT and LWCT (4.44% and 3.85%, respectively) were not significant (*p* = 0.309). 

### 3.2. Plasma Antioxidant Capacity

The effect of dietary GA on plasma antioxidant status of high and low weaning weight piglets is shown in Table 3. Although there were no statistical differences in plasma MDA content, the piglets, particularly the LW piglets fed GA numerically, reduced the MDA content in plasma on days 14 and 42. The HW piglets had higher plasma SOD activity on day 42 (*p* = 0.043), and GSH-Px activity on day 14 (*p* = 0.005) and day 42 (*p* = 0.012) compared to LW piglets, respectively (Table 3). However, there was found to be no significant GA effect or interaction between weaning weight and dietary GA on GSH-Px activity in plasma of piglets.

## 4. Discussion

The objective of the study was to evaluate the effect of dietary GA on growth performance, diarrhea incidence, and plasma antioxidant status of piglets with high and low weaning weight. Weaning is a serious period that results in low growth rate and intestinal disorders, causing diarrhea [15] and oxidative stress [20]. During this time, weaning weight and dietary composition play key factors in influencing the growth and health of piglets. Previous studies indicate that HW piglets usually go together with a higher growth rate and ADFI during the nursery period [21]. Cabrera et al. found that ADG and BW increased linearly with the increasing weaning weight [22]. In our study, we observed the same results, mainly that HW piglets had a higher BW, ADFI and ADG (except days 0–14) than LW piglets during the trial. Usually, HW piglets show a better immunity, intestinal barrier function, and absorption, which contributes to an easier adaptation to the changes caused by weaning [23]. Interestingly, our study observed that diets with GA positively affected ADG from day 0 to 42, which was mainly induced by LW piglets showing a higher BW value on day 42. No differences were found in diarrhea incidence between treatments, but the LW piglets fed GA did have the lowest diarrhea prevalence. These findings may indicate that GA promotes the growth and slightly decreases the diarrhea of LW piglets. Weaning diarrhea is associated with an inflammatory response [24] which is triggered by an increased transcription of the NF-κB signal pathway [25]. One study found that GA can suppress the activity of NF-κB and inhibit the intestinal inflammation, and finally, results in lower diarrhea incidence [26]. A study in our laboratory also showed that GA supplementation reduced inflammatory responses by inhibiting the NF-κB signaling pathway via enhancing the expression of tight junction proteins [27]. In addition, our previous study also showed that diets with 400 mg/kg GA significantly reduced diarrhea incidence of piglets but with no effects on growth performance [10]. It is worth noting that the piglets in our previous study had weaning weights that were close to those of the HW weaned piglets in this current study, illustrating that GA may be more effective to improve the growth performance of LW weaned piglets.

In the present study, the antioxidant capacity of HW piglets was significantly improved, which is in accordance with the improvement of growth performance in HW piglets. Low birth and weaning weight usually has a significant decrease in the antioxidant capacity compared to the normal weight piglets [28]. The antioxidant activity of GA has been demonstrated by several studies. Supplementation with 5% dietary grape pomace significantly increased the antioxidant activity by enhancing the SOD activity in the liver, spleen, and kidneys of weaned piglets with an initial BW at 10.70 ± 0.8 kg [29]. Diets supplemented with GA at 50 mg/kg had positive effects on meat antioxidant capacity of finishing pigs [30]. In our study, dietary GA numerically decreased MDA content in plasma while no dietary effects were observed in SOD and GSH-Px activities, which was in agreement with the results of our previous study that there were no significant improvements in the antioxidant ability of weaned piglets [10]. We speculate that the inconsistency between our experiments and other findings may be attributed to the source of GA, target organ of piglets, growth stage of pigs, and farm conditions. However, our current study suggests that GA might have a better effect on the antioxidant capacity in LW piglets, which is consistent with the specific effect on the growth performance of LW weaned piglets. The observations in this study have implications in developing new strategies to rescue the weak piglets and consequently increase the benefits to the farm. Although our previous study investigated the effect of three different dosages of GA on growth and gut health of weaned piglets, it is worth evaluating other doses of GA, especially for LW piglets in further studies.

## 5. Conclusions

In this study, we observed that HW weaned piglets showed better growth performance and systemic antioxidant capacity than LW weaned piglets, while dietary GA supplemented at 400 mg/kg had positive effects on growth performance and diarrhea incidence, particularly in LW weaned piglets.

## Figures and Tables

**Figure 1 animals-11-03323-f001:**
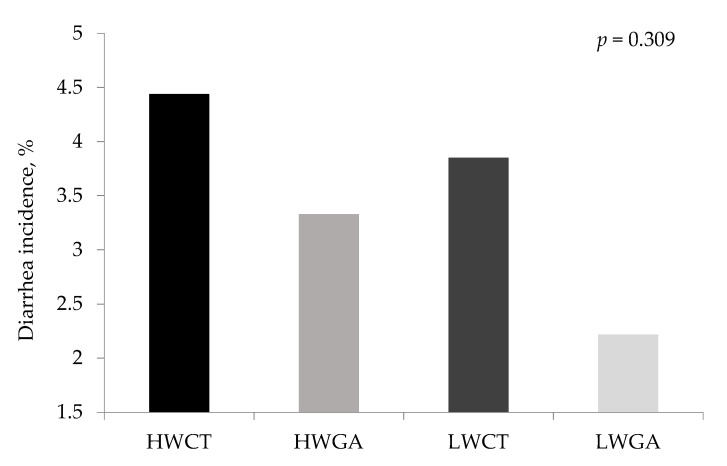
Effect of dietary gallic acid on diarrhea incidence of high and low weaning weight piglets from day 1 to day 14 post weaning.

**Table 1 animals-11-03323-t001:** Ingredient and nutrient composition of the basal diet (as fed basis).

Items	Pre-Starter (Day 0–14)	Starter (Day 14–42)
Ingredients, %		
Extruded corn	46.20	60.17
Soybean meal, 46% CP	14.60	17.50
Extruded soybean	11.50	5.00
Fish meal	5.00	3.00
Dried whey	15.00	5.00
Bran	2.842	4.142
Soybean oil	1.00	1.20
CaH_2_PO_4_	0.40	0.50
Limestone	0.80	1.00
NaCl	0.30	0.30
Choline chloride, 60%	0.05	0.05
L-Lysine H_2_SO_4_, 52.4%	1.20	1.08
DL-Methionine, 98.5%	0.09	0.08
L-Threonine, 98.5%	0.27	0.24
L-Tryptophan, 98.5%	0.02	0.01
Phytase	0.02	0.02
Acidifier	0.20	0.20
Butyric acid	0.15	0.15
Flavour	0.05	0.05
Ethoxyquin	0.02	0.02
Vitamin premix ^1^	0.048	0.048
Trace mineral premix ^1^	0.20	0.20
Total	100.00	100.00
Analyzed nutrient content		
Crude protein, %	19.43	17.69
Calcium, %	0.75	0.66
Phosphotus, %	0.66	0.61
Calculated nutrient content		
ME, kcal/kg	3400	3350
Lysine, %	1.30	1.15
Methionine, %	0.38	0.34
Threonine, %	0.76	0.68
Tryptophan, %	0.21	0.19

^1^ The premix provided the following per kg of diets: niacin, 38.4 mg; calcium pantothenate, 25 mg; folic acid, 1.68 mg; biotin, 0.16 mg; vitamin A, 35.2 mg; vitamin B_1_, 4 mg; vitamin B_2_, 12 mg; vitamin B_6_, 8.32 mg; vitamin B_12_, 4.8 mg; vitamin D_3_, 7.68 mg; vitamin E, 128 mg; vitamin K_3_, 8.16 mg; copper (CuSO_4_ · 5H_2_O), 125 mg; zinc (ZnSO_4_· H_2_O), 110 mg; selenium (Na_2_SeO_3_), 0.19 mg; iron (FeSO_4_ · H_2_O), 171 mg; cobalt (CoCl_2_), 0.19 mg; manganese (MnSO_4_·H_2_O), 42.31 mg; iodine (Ca(IO_3_)_2_), 0.54 mg.

**Table 2 animals-11-03323-t002:** Effect of dietary GA on growth performance of high and low weaning weight piglets.

	Treatment					Weight (W)			Diet (D)			*p*-Value		
	HWCT	HWGA	LWCT	LWGA	SEM	HW	LW	SEM	CT	GA	SEM	W	D	W×D
BW, kg														
Day 0	8.49	8.49	5.46	5.45	0.24	8.49	5.45	0.15	6.97	6.97	0.70	<0.001	0.977	0.973
Day 14	10.80	11.33	7.73	7.80	0.24	11.07	7.77	0.18	9.27	9.57	0.76	<0.001	0.321	0.435
Day 28	15.42	16.03	10.93	12.13	0.52	15.73	11.53	0.41	13.17	14.08	1.00	<0.001	0.140	0.607
Day 42	23.84	24.53	17.36	19.10	0.50	24.19	18.23	0.42	20.60	21.82	1.37	<0.001	0.045	0.334
ADG, g														
Day 0–14	165	203	162	168	28	184	165	19	164	186	20	0.554	0.498	0.618
Day 14–28	330	336	228	310	25	333	269	22	279	323	22	0.057	0.170	0.226
Day 28–42	602	607	460	498	30	604	479	20	531	552	34	0.004	0.502	0.613
Day 0–42	366	382	283	325	12	374	304	11	325	354	18	<0.001	0.049	0.341
ADFI, g														
Day 0–14	318	323	265	264	21	320	264	13	291	293	18	0.031	0.932	0.867
Day 14–28	660	555	420	499	38	608	460	33	540	527	48	0.013	0.791	0.086
Day 28–42	1017	988	731	885	34	1002	808	37	874	936	50	0.002	0.186	0.065
Day 0–42	665	622	472	549	28	644	511	24	569	586	37	0.004	0.616	0.105
G:F ratio														
Day 0–14	0.53	0.62	0.61	0.64	0.08	0.57	0.63	0.05	0.57	0.63	0.06	0.537	0.510	0.707
Day 14–28	0.51	0.62	0.54	0.62	0.05	0.56	0.58	0.04	0.52	0.62	0.04	0.805	0.135	0.819
Day 28–42	0.59	0.62	0.63	0.56	0.03	0.61	0.60	0.02	0.61	0.59	0.02	0.750	0.510	0.182
Day 0–42	0.56	0.62	0.60	0.59	0.03	0.59	0.60	0.02	0.58	0.61	0.02	0.770	0.370	0.256

BW = body weight; ADG = average daily gain; ADFI = average daily feed intake; G:F = gain:feed ratio; HWCT = high weight without product; LWCT = low weight without product; HWGA = high weight with 400 mg/kg GA; LWGA = low weight with 400 mg/kg GA.

**Table 3 animals-11-03323-t003:** Effect of dietary GA on plasma antioxidant status of high and low weaning weight piglets.

	Treatment					Weight (W)			Diet (D)			*p*-Value		
	HWCT	HWGA	LWCT	LWGA	SEM	HW	LW	SEM	CT	GA	SEM	W	D	W×D
MDA, mg/mL														
Day 14	1.13	0.95	1.21	0.95	0.20	1.04	1.08	0.15	1.17	0.95	0.14	0.858	0.315	0.858
Day 42	2.13	1.95	5.42	2.72	0.78	2.04	4.07	0.70	3.78	2.34	0.75	0.129	0.275	0.337
SOD, U/mL														
Day 14	16.77	16.44	17.20	17.47	0.63	16.61	17.34	0.44	16.99	16.96	0.44	0.272	0.965	0.643
Day 42	19.44	20.97	16.90	18.29	1.20	20.21	17.60	0.84	18.17	19.63	0.90	0.043	0.240	0.954
GSH-Px, U/mL														
Day 14	568	585	500	493	23	577	496	16	534	539	21	0.005	0.840	0.651
Day 42	564	564	484	487	28	564	485	19	524	525	22	0.012	0.954	0.973

MDA = malondialdehyde; SOD = superoxide dismutase; GSH-Px = glutathione peroxidase; HWCT = high weight without product; LWCT = low weight without product; HWGA = high weight with 400 mg/kg GA; LWGA = low weight with 400 mg/kg GA.

## Data Availability

The data presented in this study are available on request from the corresponding author.

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
