# Peer review of "The Influence of Dietary Gallic Acid on Growth Performance and Plasma Antioxidant Status of High and Low Weaning Weight Piglets"

_animals, 2021, doi:10.3390/ani11113323_

Round 1
Reviewer 1 Report
The paper is relatively straight forward and relatively well written.
A few comments and suggestions:
- I think in the summary, the factor of weaning weight needs to be mentioned because the experiment is a 2 x 2 factorial
- In the Materials and method, define the weaning weights that are considered high (such as weight above 8 kg) or low (weights below 6 kg as an example)
- Needs spelling check (line 162)
- Line 148 , high or low W fed with GA demonstrated a numerical reduction in diarrhea incidence It may be good to mention that as well.
- Line 185-186,"...which was mainly induced by LW piglets showing a higher BW 186 value on day 42". Higher than what?
Author Response
The paper is relatively straight forward and relatively well written.
Response (R) to Reviewer 1: Thank you very much for your kind comments.
A few comments and suggestions:
1. I think in the summary, the factor of weaning weight needs to be mentioned because the experiment is a 2 x 2 factorial
R: We added the range of weight in the abstract, please see Line 25.
2.In the Materials and method, define the weaning weights that are considered high (such as weight above 8 kg) or low (weights below 6 kg as an example)
R: Thank you for the suggestion, we added the weight with the variation of the piglets. Please see line 84.
3. Needs spelling check (line 162)
R: We modified the typo. Please see Line 172.
4. Line 148 , high or low W fed with GA demonstrated a numerical reduction in diarrhea incidence It may be good to mention that as well.
R: We added the description according to your suggestion. Please see Line 158-160.
5. Line 185-186,"...which was mainly induced by LW piglets showing a higher BW value on day 42". Higher than what?
R: We added the information. Please see Line 192.
Reviewer 2 Report
General Comments:
The authors investigated the effects of dietary gallic acid as a natural antioxidant on the performance, diarrhea and plasma parameter of nursery pigs with high and low weaning weights. It is interesting that dietary gallic acid promoted body weight gain of piglets particularly with low weaning weights, which is reasonable. The justification and the hypothesis for the experiment are acceptable and the experimental procedure is solid. The manuscript reads well. However, my major concern on this work is the use of a single-phase diet for 42 days of nursery period which is very uncommon in practice. This shortcoming need to be mentioned in the manuscript possibly in the discussion section.
Specific Comments
L 20-21: provide the initial body weights
L 22: … factorial treatment arrangement comparing …
L 50, 76: Here and other places, please avoid an abbreviation at the beginning of a sentence
L 70-71: Provide the protocol approval number.
L 77-80: Please provide the body weights for low and high weaning weights.
L 83: … factorial treatment arrangement comparing …
Table 1: L-Lys-HCl, 78.8% ??
DL-Met, 98.5%??
L-Thr
L-Trp …
Please clarify these
What was the antioxidant used?
A superscript number 1 should also be next to “Vitamin premix”
ME, kcal/kg
L 139: Here and in other result descriptions, delete “significantly” or “significant” as the the statistical significant has been readily defined in the stats section.
L 147: Delete “(p > 0.05)” which is redundant as the statistical significant has been readily defined in the stats section.
Table 2: The super script letters in the results are redundant as there are only two treatments for main effect comparisons. The footnote also needs to be deleted.
Figure 1: Indicate variability and stats for the comparisons for the diarrhea incidence.
L 162: Delete “(p > 0.05)” which is redundant as the statistical significant has been readily defined in the stats section.
Table 3: The superscript letters in the results are redundant as there are only two treatments for main effect comparisons. The footnote also needs to be deleted.
L 172: The footnote for superscript letters also needs to be deleted.
L 180: Here and throughout the manuscript, avoid abbreviations at the beginning of a sentence.
L 201-217: For the inconsistency between the present work and previous experiments may need to be explained at least partially or speculated.
Author Response
The authors investigated the effects of dietary gallic acid as a natural antioxidant on the performance, diarrhea and plasma parameter of nursery pigs with high and low weaning weights. It is interesting that dietary gallic acid promoted body weight gain of piglets particularly with low weaning weights, which is reasonable. The justification and the hypothesis for the experiment are acceptable and the experimental procedure is solid. The manuscript reads well. However, my major concern on this work is the use of a single-phase diet for 42 days of nursery period which is very uncommon in practice. This shortcoming need to be mentioned in the manuscript possibly in the discussion section.
Response (R) to Reviewer 2: We would highly appreciate the comments provided the reviewer to improve the quality of our manuscript. We apologize that we did not write clearly about the diet and indicate the stages in the Table 1. Actually, two diets were designed according to the NRC 2012 nutrient requirements for the piglets during the trial, for the pre-starter period from 0 to 14 day and starter period from 14 to 42 day of trial. Please see Line 90-94 and Table 1.
Specific Comments
L 20-21: provide the initial body weights
R: Thank you for the suggestion, and we have added the information. Please see Line 25.
L 22: … factorial treatment arrangement comparing …
R: We modified the sentence. Please see Line 24.
L 50, 76: Here and other places, please avoid an abbreviation at the beginning of a sentence
R: We rewrote the relevant sentences. Please see Lines 45, 53-54, 79-80, 251.
L 70-71: Provide the protocol approval number.
R: Please see Line 75.
L 77-80: Please provide the body weights for low and high weaning weights.
R: We added the weight with the variation of the piglets. Please see Line 84.
L 83: … factorial treatment arrangement comparing …
R: Please see Line 88.
Table 1: L-Lys-HCl, 78.8% ??
DL-Met, 98.5%??
L-Thr
L-Trp …
Please clarify these
What was the antioxidant used?
R: Thank you for your reminder, we have added the content in the Table 1.
A superscript number 1 should also be next to “Vitamin premix”
ME, kcal/kg
R: Please see Table 1.
L 139: Here and in other result descriptions, delete “significantly” or “significant” as the the statistical significant has been readily defined in the stats section.
R: We deleted the words in the Results section according to your suggestion.
L 147: Delete “(p > 0.05)” which is redundant as the statistical significant has been readily defined in the stats section.
R: We deleted the indication.
Table 2: The super script letters in the results are redundant as there are only two treatments for main effect comparisons. The footnote also needs to be deleted.
R: We deleted the superscripts and footnote. Please see Table 2.
Figure 1: Indicate variability and stats for the comparisons for the diarrhea incidence.
R: We added the P value in the Figure 1.
L 162: Delete “(p > 0.05)” which is redundant as the statistical significant has been readily defined in the stats section.
R: We deleted the indication.
Table 3: The superscript letters in the results are redundant as there are only two treatments for main effect comparisons. The footnote also needs to be deleted.
R: We deleted the superscripts and footnote. Please see Table 3.
L 172: The footnote for superscript letters also needs to be deleted.
R: Thank you for the suggestion, we delete it.
L 180: Here and throughout the manuscript, avoid abbreviations at the beginning of a sentence.
R: We modified the sentence. Please see Line 189.
L 201-217: For the inconsistency between the present work and previous experiments may need to be explained at least partially or speculated.
R: We modified the sentence. Please see line 214-229.
Reviewer 3 Report
Different feed additives could improve pig performance or immune function. Study on various feed additives that can boosting the piglets' immune system and growth potential as well as reduce negative impact of weaning is a very important area of interest for scientists and breeders, especially when we want to reduce antibiotic usage from one side and we have more numerous litters what is connected with higher weaning weight differentiation. The topic chosen by the Authors fits in with this area, but the manuscript requires some changes and explanations.
It is commonly known that weaning weight influence daily gains and health status after weaning so this part of the study aim should be removed, especially when we have so small experimental population. Of course in summary you can conclude add that obtained results confirm the commonly known rules. Your aim was rather to investigate the influence of GA additive on ...., try to clarify it.
Because in your earlier studies GA additive was shown to positively affect the piglets' intestinal morphology and the occurrence of diarrhea add in introduction something more why you want to investigate it more, what distinguishes it from other additives of this type - availability, price?
Present your methodology more carefully:
- present the range for high and low body weight
- were these piglets reared at the same farm where they were born?, it is why you didn't use adaptation period?
- experimental period was 42 days, I couldn't find in Table 1 dietary phases and their duration, was it 2 week for every feed mixture? How did the feed changes look like, whether was it a sudden change overnight?
- when/what way the GA was added to feed mixture?
- it will be better to present results only for 4 experimental groups to show influence of GA additive. I don't know the price of GA, it could be interesting if it is worth to use it for HW piglets.
Differences between LW and HW groups are rather expected and these results do not add anything, the same for CT and GA.
- instead of neck vein you should write vena jugularis externa
Author Response
Different feed additives could improve pig performance or immune function. Study on various feed additives that can boosting the piglets' immune system and growth potential as well as reduce negative impact of weaning is a very important area of interest for scientists and breeders, especially when we want to reduce antibiotic usage from one side and we have more numerous litters what is connected with higher weaning weight differentiation. The topic chosen by the Authors fits in with this area, but the manuscript requires some changes and explanations.
It is commonly known that weaning weight influence daily gains and health status after weaning so this part of the study aim should be removed, especially when we have so small experimental population. Of course in summary you can conclude add that obtained results confirm the commonly known rules. Your aim was rather to investigate the influence of GA additive on ...., try to clarify it. Because in your earlier studies GA additive was shown to positively affect the piglets' intestinal morphology and the occurrence of diarrhea add in introduction something more why you want to investigate it more, what distinguishes it from other additives of this type - availability, price?
Responses (R) to Reviewer 3: We appreciate the reviewer’s efforts for improving the quality of our manuscript. Our experiment was designed as a 2 × 2 factorial treatment arrangement comparing different weaning weights and dietary GA, that’s why we illustrate the influence of weaning weight to piglets. We agree that weaning weight apparently influence the growth and health status of weaned piglets. Although the observations in our previous study demonstrated that supplementation of 400 mg/kg GA could improve the gut health and reduce the diarrhea incidence of weaned piglet, the positive effect on growth performance to weaned piglets was not observed. Thus, this study was conducted to investigate whether GA has specific effect on growth performance of low weaning weight piglets. In addition, our study could enforce the application of gallic acid to be a potential alternative of antibiotics in pig production.
Present your methodology more carefully:
- present the range for high and low body weight
R: We added the weight with the variation of the piglets. Please see Line 83.
- were these piglets reared at the same farm where they were born?, it is why you didn't use adaptation period?
R: Yes, all the piglets were reared from the same batch of 319 piglets from the same farm (line 84-85), so we didn’t arrange the adaptation period.
- experimental period was 42 days, I couldn't find in Table 1 dietary phases and their duration, was it 2 week for every feed mixture? How did the feed changes look like, whether was it a sudden change overnight?
R: We did not write clearly about the dietary phases, the pre-starter period from 0 to 14 day and starter period from 14 to 42 day of trial. we add the specific application time of different diets in Table 1. In addition, in the animals and experimental design section, we also illustrate it, please see line 92-94. On the morning day 14 of the trial, the pre-starter food was collected and the starter food was afforded to piglets.
- when/what way the GA was added to feed mixture?
R: We added the GA to the vitamin and mineral complexes and mixed by hand, and then this mixture was added to feed mixed by machine.
- it will be better to present results only for 4 experimental groups to show influence of GA additive. I don't know the price of GA, it could be interesting if it is worth to use it for HW piglets.
R: We agree on your suggestion that mainly show the influence of GA additive, while we also would like to investigate whether GA has improvement on piglets with both weights or specifically affect the weak piglets. Concerning the price of GA (70-100 RMB/kg), it is worth to use it for HW piglets using the dosage in our study as the alternative of antibiotics.
Differences between LW and HW groups are rather expected and these results do not add anything, the same for CT and GA.
R: In the present study, we got the same results that high weight piglets had better growth performance, lower diarrhea incidence and better plasma antioxidant capacity. High weight piglets were easy adapted to weaning challenges. This is consistent with previous studies. Piglets fed GA had better performance than the control piglets. However, it is worth to note that dietary GA supplementation had better effects to low weight piglets, which may decrease the negative effects caused by weight of piglets weaned at the same batch of the farm. In addition, the interactions between weaning weight and dietary GA also had minor effects on piglets.
- instead of neck vein you should write vena jugularis externa
R: Thank you for the suggestion. Please see Line 122.
Round 2
Reviewer 3 Report
Still I'm not satisfied from your changes. I'm sure you have to change your aim or formulate it differently. Reading the aim we have to see the need to study a presenting problem. so why you can't write that you want to investigate if weaning weight would increase growth etc.... - we know it! I expect that your goal was to check if additive of GA can influence on health status and production results regardless of whether it is administered to heavy or light piglets. But I wanted you to do some changes/modifications in your aim (the first part).
l.66 - correct this sentence: "So, we hypothesized that growth performance and diarrhea after weaning could be improved by supplementing the diet with 400 mg/kg GA to weaned piglets..."
l.78 - double bracket
This sentence from your answer should be included in Materials:
"On the morning day 14 of the trial, the pre-starter food was collected and the starter food was afforded to piglets" (By the way we don't recommend so sudden feed changes ;)).
And the next : " We added the GA to the vitamin and mineral complexes and mixed by hand, and then this mixture was added to feed mixed by machine" - it is your method.
And the last part of my reservations connected with the scheme of experiment which I mentioned earlier.
l.147 - we don't observe decrease in BW and ADFI in the LW piglets, just these piglets grew more slowly (what is normal and expected).
l.149-152 - we can find that piglets feed GA showed higher BW compared to control ones but this 1,22 kg more results from higher BW of HWGA piglets! You can't explain it by the supplementations of GA for lighter piglets. - back to it in the discussion.
l.158-150 - you add that diarrhea incidents were reduced by 25 and 42% respectively in HWGA,LWGA and HWCT,LWCT groups. These percentages could impressed but when we look at numbers of these incidents they are small. So keep scientific objectivity in presenting the results. Results are results!, just present them.
- 173 - change capital letter Effects
- 191 - don't repeat the sentence from the Introduction
l.212 - I don't understand - to which group: HW or GA?
l.217 - will be better: "Low birth and weaning weight usually ...." and probably" has" not "had"
In Conclusion or in the end of Discussion: why don't you suggest that it is worth to evaluate other doses of GA especially for LW piglets? - don't you think it worth further studies?
Author Response
Still I'm not satisfied from your changes. I'm sure you have to change your aim or formulate it differently. Reading the aim we have to see the need to study a presenting problem. so why you can't write that you want to investigate if weaning weight would increase growth etc.... - we know it! I expect that your goal was to check if additive of GA can influence on health status and production results regardless of whether it is administered to heavy or light piglets. But I wanted you to do some changes/modifications in your aim (the first part).
Responses (R) to Reviewer 3: We appreciate your feedback and comments aimed to improve the quality of our manuscript. Of course, we completely agree on your viewpoint that a new study should be conducted to solve a presenting problem. Thus, we modified the title and the aim of our manuscript according to your suggestion. Please see Line 21-23 and 64-71.
l.66 - correct this sentence: "So, we hypothesized that growth performance and diarrhea after weaning could be improved by supplementing the diet with 400 mg/kg GA to weaned piglets..."
R: We modified the sentence. Please see Line 65-67.
l.78 - double bracket
R: We modified them. Please see Line 78-79.
This sentence from your answer should be included in Materials:
"On the morning day 14 of the trial, the pre-starter food was collected and the starter food was afforded to piglets" (By the way we don't recommend so sudden feed changes ;)).
R: We added the sentence in the M&M section. Please see line 95-97. (We agree, usually the change of diet produces the stress and easily results in diarrhea particular in the weak piglets, while it would be worth to evaluate if the additive could attenuate the stress during the sudden diet change in the experimental trial)
And the next : " We added the GA to the vitamin and mineral complexes and mixed by hand, and then this mixture was added to feed mixed by machine" - it is your method.
R: We added the sentence in the M&M section. Please see Line 94-95.
And the last part of my reservations connected with the scheme of experiment which I mentioned earlier.
R: Thank you very much for your suggestion and we totally agree with you. Firstly, we changed our title and objective of the study to avoid the sense of weaning weight; Secondly, we also modified the interpretation in the Results section and description of the Tables. However, the design of our study would be better to keep as 2 × 2 factorial study, because the weaning weight is an essential factor for our study.
l.147 - we don't observe decrease in BW and ADFI in the LW piglets, just these piglets grew more slowly (what is normal and expected).
R: We agree that LW piglets undoubtedly grow slowly and have lower ADFI compared to HW piglets. Thus, we delete this sentence to avoid misleading.
l.149-152 - we can find that piglets feed GA showed higher BW compared to control ones but this 1,22 kg more results from higher BW of HWGA piglets! You can't explain it by the supplementations of GA for lighter piglets. - back to it in the discussion.
R: Thank you for your reminder. In the Table 2, we can see that the final BW in GA treatments increased by 1.22 kg (21.82-20.60 = 1.22 kg) compared to the CT group. Compared to the HWCT group, the BW in the HWGA group increased by 0.69 kg (24.53-23.84 = 0.69 kg), while the BW in the LWGA group increased by 1.74 kg (19.10-17.36 = 1.74kg) compared to the LWCT group. Thus, we explained the final BW increased by the supplementations of GA for lighter piglets due to the increasement of 1.74 kg in the LW piglets is higher than 0.69 kg in the HW piglets.
l.158-150 - you add that diarrhea incidents were reduced by 25 and 42% respectively in HWGA,LWGA and HWCT,LWCT groups. These percentages could impressed but when we look at numbers of these incidents they are small. So keep scientific objectivity in presenting the results. Results are results!, just present them.
R: We modified the description according to your suggestion. Please see Line 159-163.
- 173 - change capital letter Effects
R: Please see Line 173.
- 191 - don't repeat the sentence from the Introduction
R: We removed the sentence and modified the content in the Introduction section. Please see Line 57-61.
l.212 - I don't understand - to which group: HW or GA?
R: The weaning weight (8.40 ± 0.09 kg) in our previous study was close to the HW groups (8.49 ± 0.18 kg) in our current study, and we modified the sentence to avoid the misleading. Please see Line 211-214.
l.217 - will be better: "Low birth and weaning weight usually ...." and probably" has" not "had"
R: Thank you for the revised suggestion. Please see Line 217.
In Conclusion or in the end of Discussion: why don't you suggest that it is worth to evaluate other doses of GA especially for LW piglets? - don't you think it worth further studies?
R: Thank you for your suggestion. Please see Line 233-235.